# Augmentation of Growth Hormone by Chewing in Females

**DOI:** 10.3390/nu15163628

**Published:** 2023-08-18

**Authors:** Emi Okamura, Kaori Ikeda, Fumika Mano-Usui, Sachiko Kawashima, Aki Kondo, Nobuya Inagaki

**Affiliations:** 1Department of Diabetes, Endocrinology and Nutrition, Graduate School of Medicine, Kyoto University, Kyoto 606-8507, Japan; e_okamura@kuhp.kyoto-u.ac.jp (E.O.);; 2Department of Clinical Research Facilitation, Institute for Advancement of Clinical and Translational Science, Kyoto University Hospital, Kyoto 606-8507, Japan; 3Preemptive Medicine and Lifestyle Related Disease Research Center, Kyoto University Hospital, Kyoto 606-8507, Japan; 4Medical Research Institute KITANO HOSPITAL, P.I.I.F. Tazuke-Kofukai, Osaka 530-8480, Japan

**Keywords:** growth hormone, ghrelin, chewing, muscle mass

## Abstract

Sarcopenia is an age-related condition characterized by progressive loss of muscle mass and strength. Age-related decline in the secretion of growth hormone (GH), a condition called somatopause, is thought to play a role in sarcopenia. As pharmacological GH has adverse effects, we attempted to increase physiological GH. While the relationship between chewing and ghrelin levels has been studied, there are no reports on the relationship between chewing and GH. The aim of this study was to clarify the effects of chewing on the muscle anabolic hormones serum GH and plasma ghrelin. Thirteen healthy adults ingested a chewy nutrition bar containing 5.56 g of protein, 12.71 g of carbohydrate, and 0.09 g of fat on two different days, chewing before swallowing in one trial and swallowing without chewing in the other. Blood samples were taken before and after ingestion (0, 15, 30, and 60 min); GH, acylated ghrelin, glucose, insulin, amino acids, and lactate were measured. Two-way repeated ANOVA revealed a significant difference in the GH concentrations between the “Chew trial” and “Swallow trial” in females (*p* = 0.0054). However, post-hoc analyses found no statistically significant difference at each time point. The area under the curve of the percentage increase in GH was significantly increased in the “Chew trial” compared with the “Swallow trial” in females (12,203 ± 15,402% min vs. 3735 ± 988% min, *p* = 0.0488). Chewing had no effect on glucose, insulin, amino acids, or lactate concentrations. Thus, we found that chewing a protein supplement rather than swallowing it without chewing elevates the blood GH concentration. These results serve as a rationale for larger research and longitudinal studies to confirm the impacts of chewing on GH secretion.

## 1. Introduction

Sarcopenia is an age-related condition characterized by progressive loss of muscle mass and strength. Sarcopenia increases the risk of falls and fractures [1], and can therefore lead to poor quality of life (QOL) [2], long-term care placement [3], and death [4]. The primary cause of sarcopenia is age-related physiological change; secondary causes include nutritional deficiency, inactivity, organ dysfunction, inflammation, malignancy, and endocrinological disturbances [5]. Age-related endocrine-associated changes occur in androgen, vitamin D, and the GH IGF-1 axis. Age-related decline in the secretion of growth hormone (GH) [6], a condition called somatopause, plays a role in both primary and secondary sarcopenia. Since GH is a strong anabolic agent in muscle, GH treatment increases lean body mass [7,8,9,10] and decreases relative adipose-tissue mass [9,10]. The known side effects of pharmacological GH, however, are fluid retention and insulin resistance, which are dose dependent.

Physiologically, GH release is mainly controlled by growth hormone-releasing hormone (GHRH) and its suppressing hormone, somatostatin. While most GH outputs occur during sleep in males, the GH secretory pulses in the daytime are more frequent in females, which is correlated with circulating free estradiol levels [11,12]. In starvation, ghrelin, a gastrointestinal peptide hormone released from the endocrine cells in the stomach, is another secretagogue of GH. Ghrelin circulates in both acylated and non-acylated forms. Non-acyl ghrelin is generated by the deacylation of ghrelin in plasma [13]. Data on the physiological interactions between acylghrelin and non-acylghrelin are limited, but the administration of acylghrelin causes an increase in non-acylghrelin, whereas the administration of non-acylghrelin does not alter plasma acylghrelin [14]. The acylated form of ghrelin is the major active form of human ghrelin and stimulates GH release through its receptor, growth hormone secretagogue receptor type 1a (GHSR-1a); the non-acylated form does not bind or activate GHSR-1a. Acylated ghrelin independently increases food intake and lean body mass [15]; it is elevated in blood in the fasting state and is suppressed in response to food intake. The total plasma ghrelin concentration is significantly increased by sham feeding in which subjects chew but do not swallow a test meal [16]. In a study of patients with colorectal cancer, gum chewing was found to significantly increase the serum level of non-acylated ghrelin [17]. However, there are few reports on the association of chewing with acylated ghrelin.

Amino acids are a nutritional secretagogue of GH; arginine, lysine, and ornithine have been investigated for their effect on GH release. Indeed, arginine is often used as a stimulator of GH secretion as a diagnostic test for GH deficiency. Arginine partly increases GH release by suppressing endogenous somatostatin secretion [18]; it also increases calcium influx in pituitary cells, suggesting a stimulatory effect on GH secretion [19]. Arginine or other amino acids can stimulate GH release by both intravenous and oral administration. Intravenous infusion of 183 mg of arginine per kilogram of body weight was found to increase plasma GH over 20-fold in females [20] and infusion of 30 g of arginine to elevate serum GH concentrations 8.6-fold in males [18]. In addition, glucose is known to rapidly decrease the GH concentration [21].

Physical exercise is also a potent stimulator of GH release [22]; GH secretion linearly increases with the intensity of exercise [23]. Although the mechanism of GH secretion in response to exercise has not been elucidated, several related factors have been identified: direct neural stimulation of the anterior pituitary, secretion of nitric oxide, an increase in circulating catecholamines and lactate, and a decrease in pH. Lactate is a major factor in the promotion of GH secretion; muscle fatigue during high-intensity exercise is associated with the accumulation of lactate and a concomitant lowering of pH in muscle. GH is very sensitive to changes in pH and the blood lactate level; indeed, a single injection of sodium lactate evokes a significant increase in the serum GH concentration in rats [24]. Masticatory movement also causes muscle fatigue similar to that of movements of limb muscles [25].

The chewing ingestion of protein supplements involves all these potent factors for GH stimulation. While chewing itself can increase the ghrelin level and GH can be elevated by increased plasma ghrelin, there have been no previous reports on the relationship between chewing and GH concentration. The aim of this study was to examine the effect of chewing on circulating serum GH and plasma ghrelin levels using a nutrition bar.

## 2. Materials and Methods

### 2.1. Participants

Healthy men and women were recruited between March 2019 and March 2020. None of the participants had diabetes or gastric/intestinal/pancreatic disease. No medications known to affect GH, ghrelin, glucose, insulin, amino acids, or lactate were taken. We recruited subjects who did not regularly take medication. The participants were instructed to abstain from supplements during the trial. None of the participants were pregnant or were night workers. All participants gave written, informed consent for participation in the study before protocol-specific procedures were carried out. The protocol of the study was approved by the Kyoto University Graduate School and Faculty of Medicine, Ethics Committee. The research is registered at the University Hospital Medical Information Network (UMIN) Clinical Trials Registry (UMIN000034542).

### 2.2. Study Design

Participants were individually allocated in a cross-over design to two ingestion tests, “Chew” and “Swallow” (without chewing) a nutrition bar on different days. The participants were not informed in advance regarding the sequence of the “Chew trial” and “Swallow trial” until just before the test. The order was decided according to the convenience of preparation of the nutrition bar. All ingestion tests started at 9:00–10:00 a.m. after 10-h overnight fasting and refraining from taking caffeine (water allowed). In the “Chew trial”, participants chewed a single nutrition bar at least 300 times within 4 min together with about 180 mL water; “300 times in 4 min” was determined to be the appropriate frequency for ingesting this nutrition bar with thorough chewing. Because the bar dissolves during chewing, it was not possible to chew it more than 300 times. In the “Swallow trial”, the participants ingested a single, finely chopped nutrition bar without chewing within 4 min with a similar amount of water. As GH is stimulated by estrogen and its concentration changes during the menstrual cycle, all tests were scheduled to avoid the single week after the onset of participant menses.

Blood samples were collected before and after ingestion (0, 15, 30, and 60 min); serum GH, plasma acylated ghrelin, serum glucose, serum insulin, plasma amino acids, and lactate levels were measured. Participants were not allowed other food or exercise during the 60-min test.

### 2.3. Test Meal

The nutrition bar was designed to have a texture feasible for both thorough chewing or swallowing after being finely chopped and contained as much protein as possible and little sucrose and fat. It contained 5.56 g of protein, 12.71 g of carbohydrate, and 0.09 g of fat. The protein was mainly derived from milk protein and was enriched by leucine. The total amount of leucine was 1.37 g. The carbohydrate consisted of 5.56 g carbohydrate from reduced sugar syrup (a mixture of sugar alcohols from syrup obtained by the hydrolyzation of starch using acid or enzymes and reduction), 3.55 g from palatinose, and 3.58 g from sucrose and starch syrup and contained approximately 65 kcal with 28% maltitol and 28% palatinose.

The nutrition bar (about 8 cm× 2.5 cm× 0.8 cm in size) was manufactured by UHA Mikakuto Co., Ltd. (Osaka, Japan), with a texture similar to that of dried fruit; it is dissolved with chewing according to the moisture and temperature in the mouth and has a somewhat sweet taste reminiscent of yogurt.

### 2.4. Data Collection

Venous blood was collected via a cannula in the median cubital vein. Blood samples were collected after 15 min of resting, with at least a 15-min interval from cannula placement to blood collection. Blood samples were put into blood collection tubes (one without any additives, one containing EDTA 2 Na, and one containing 0.8 N perchloric acid) and centrifuged for 10 min at 4 °C. The resultant serum and plasma were stored in appropriate tubes at −80 °C until analysis. 

Blood samples for acylated ghrelin were collected into chilled tubes containing EDTA 3 K and aprotinin (50 KIU/mL Blood) and immediately centrifuged at 1500 g for 15 min at 4 °C. Acylated ghrelin is unstable in blood and easily reverts to its non-acylated, inactive form. The separated plasma after centrifugation was acidified by 1/10 volume of 1 M HCl and stored at −80 °C until assay.

### 2.5. Measurements

Serum GH levels were determined by Electro Chemiluminescence Immunoassay using Cobas 8000 analyzer series module e801 (Roche Diagnostics, Mannheim, Germany; %CV of intra-assay variability <15%). The serum glucose level was determined by hexokinase enzymatic method (L-type Wako Glu2; Wako Pure Chemical Industries, Osaka, Japan) using LABOSPECT 008 α (HITACHI, Tokyo, Japan) and JCA-BM8060 (JEOL, Tokyo, Japan). The lactate level was determined by enzymatic method (Deteminer LA; Minaris Medical, Tokyo, Japan) using JCA-BM9130 (JEOL, Tokyo, Japan). Serum insulin levels were determined by Chemiluminescent Immunoassay using Architect i2000 SR (Abbott, Osaka, Japan). Plasma amino acids fractionation was determined by High-Pressure Liquid Chromatography using a Modification of the 6-Aminoquinolyl-N-Hydroxysuccinimidyl Carbamate (AQC) Method on a Waters Acquity UPLC system. GH, glucose, insulin, amino acids, and lactate levels were analyzed by the LSI Medience Corporation (Kyoto, Japan). Acylated ghrelin levels were measured using an Active Ghrelin ELISA Kit (# 97751, LSI Medience, Tokyo, Japan).

### 2.6. Statistical Analysis

Two-way repeated measure analysis of variance (ANOVA) was used to evaluate the differences in GH, acylated ghrelin, glucose, insulin, amino acids, and lactate concentration between the “Chew trial” and “Swallow trial” at the four time points. Post-hoc analyses of inter-trial comparison with Bonferroni correction were performed at each time point. These post-hoc analyses at each time point and the area under the curve (AUC) of percentage increase in GH were performed by pairwise comparisons using Wilcoxson’s signed rank test. Two-tailed *p* < 0.05 was considered significant. All analyses were performed using the JMP Pro 15.1.0 (SAS Institute Inc., Cary, NC, USA).

## 3. Results

A total of thirteen healthy adults (three males and ten females; age 38.2 ± 6.5 years and BMI 21.2 ± 1.6 kg/m^2^) participated in the study. None of them took oral contraceptives, and no postmenopausal females were included. All participants first performed one of the two ingestion tests (Chew trial (n = 4) or Swallow trial (n = 9)). After at least 4 days, the participants then underwent the other ingestion test in a cross-over design.

In the “Chew trial”, the serum GH level tended to increase after ingestion of the nutrition bar (Figure 1a). In the “Swallow trial”, GH tended to decrease over time. Two-way repeated ANOVA measures revealed a significant difference in GH concentration between the “Chew trial” and “Swallow trial” (*p* = 0.0118). Due to the difference in the GH secretion between males and females, the results were separately analyzed; there was a significant difference between the “Chew trial” and “Swallow trial” in females (n = 10, *p* = 0.0054) (Figure 1b), but no significant difference in males (n = 3) (Figure 1c). Post-hoc analyses revealed no significant differences between the “Chew trial” and “Swallow trial” at each time point. The AUC of percentage increase in GH was significantly larger in the “Chew trial” compared with the “Swallow trial” in females (12,203 ± 15,402% min vs. 3735 ± 988% min, *p* = 0.0488) (Figure 1d).

In females, acylated ghrelin tended to increase in the “Chew trial” and tended to decrease in the “Swallow trial”, but there was not a statistically significant difference (Figure 2a). The AUC of the percentage increase in ghrelin tended to be higher in the “Chew trial” than in the “Swallow trial,” but there was no statistically significant difference (10,331 ± 15,898% min vs. 5746 ± 6755% min, *p* = 0.0645) (Figure 2b).

The glucose and insulin level did not differ between the “Chew trial” and the “Swallow trial” by ANOVA in females (Figure 3a,b). There was also no significant difference in the AUC of the percentage increase in glucose or insulin between the “Chew trial” and the “Swallow trial” (Figure 3c,d).

Amino acids, particularly arginine, lysine, and ornithine, which are reported to be GH stimulators, did not differ between the “Chew trial” and the “Swallow trial” by ANOVA (Figure 4a–c). The lactate level showed no significant difference between the “Chew trial” and the “Swallow trial” by ANOVA (Figure 4d). There was no difference in the AUC of the percentage increase in amino acids or lactate between the “Chew trial” and the “Swallow trial” (Figure 4e–h).

## 4. Discussion

Our findings show that chewing a protein nutrition bar rather than swallowing it without chewing increased the serum GH concentration after ingestion. We speculate that chewing a protein supplement rather than just swallowing may promote GH secretion, which needs to be clarified by further study.

The relationship between mastication and ghrelin secretion has been reported in several studies, but the results are controversial. Ghrelin has been reported to be increased by chewing [16,17] and to be decreased by chewing [26,27], but these findings regard total ghrelin or des-acyl ghrelin. In one study, more chewing before swallowing pizza was found to significantly elevate postprandial insulin and glucose levels and result in a trend toward the suppression of des-acyl ghrelin [26]. Another study reported that, compared to 15 chews, 40 chews of a test meal (68% of energy as carbohydrate, 21% of energy as fat, and 11% of energy as protein) resulted in a lower postprandial total ghrelin concentration in a group of healthy, normal weight, and obese individuals [27]. Given that the active form of human ghrelin is its acylated form, which binds and activates GHSR-1a, the effects on the concentration of acylated ghrelin remain to be determined.

However, postprandial ghrelin release may be altered by the type of nutrient ingested, leading to increased plasma total ghrelin in contrast to the suppressive effect of carbohydrate and fat-rich meals [28]. Gastrin, a hormone mostly stimulated by protein intake, directly acts on ghrelin cells to stimulate ghrelin secretion resulting in an increase in the plasma acylated ghrelin concentration [29]. Gastrin is also stimulated by chewing [17,30]. In the present study, the nutrition bar contained a somewhat high percentage of energy (34%) as protein. While ghrelin generally decreases after meals, the combination of the small amount of protein intake and chewing in the current study could together contribute to the ameliorated ghrelin decrease, which might well yield an increase in serum GH. As we have no data on gastrin in the current study, this question remains.

The nutrition bar used in this study only slightly raised the blood glucose level and chewing did not affect glucose or insulin levels; maltitol, the sugar alcohol contained in the reduced syrup used in our test meal, is poorly absorbed in the small intestine. The peak concentration of blood glucose and insulin is, therefore, less after maltitol than after sucrose ingestion [31]. On the other hand, palatinose has the same composition and caloric value (4 kcal/g) as sucrose but increases blood glucose and insulin levels more slowly [32]. Glucose and insulin levels have been shown to play an important role in postprandial ghrelin suppression [33]. Glucose administration also rapidly decreases the GH concentration [21]. These initial suppressions are related to a glucose-mediated increase in hypothalamic somatostatin release [34]. There are previous reports evaluating GH suppression by glucose loading in healthy subjects that show a decrease in GH when blood glucose levels are increased to about 120–150 mg/dL [21,35,36,37]. In the current study, the smaller elevation of glucose and insulin after nutrition bar ingestion may underlie the lesser suppression of acylated ghrelin and serum GH.

Although chewing was not found to change the amino acid concentrations in this study, chewing efficiency is known to influence protein metabolism [38]. For example, in a study of healthy, elderly individuals, those with good teeth were found to experience a more rapid rise and a greater peak of plasma amino acids than denture wearers after a meat meal. This implies that poor chewing efficiency slows and worsens amino acid absorption. Poor mastication also delays the gastric emptying rate [39,40]. Denture wearers swallow fewer times during a meal than those with good teeth [41]. However, in this study, there was no difference in the concentration of amino acids between the groups. Thus, while amino acids are known to be stimulators of GH secretion, the increased serum GH level after chewing in the current study would not be attributable to amino acids.

The after-exercise level of GH secretion varies from report to report. Kanaley et al. [42] found an approximately two-fold increase in GH secretion after 30-min exercise at a constant velocity on a treadmill, which is comparable to the increase in serum GH observed in this study. A low GH concentration is associated with sarcopenia in the elderly and the GH concentration has also been found to have a positive correlation with skeletal muscle mass [43]. Since a correlation has been established between acute GH increases after resistance exercise and long-term muscle fiber type I and II hypertrophy [44], the significance of increased blood GH concentration by chewing warrants further research.

There was no significant difference in the lactate levels between the chewing and not chewing groups. Extreme chewing can provoke fatigue in the masticatory muscles [45] and induce the production and release of lactate into the blood [46]. However, the nutrition bar used in this study was manufactured to dissolve after chewing about 300 times per 4 min, making it impossible to chew to the level of intense exercise.

Our study has several limitations, including the small sample size and the young age of the participants. Possible confounding factors affecting the relationship between chewing and increased GH are the regulation mechanisms of circulating acylghrelin and non-acylghrelin, the interaction between acylgherlin and non-acylgherlin, the unclarified biological activity of non-acylghrelin, and the impact of gastrin. A similar study evaluating these factors with a larger sample including more diverse participants is warranted. Although females showed an increase in serum GH when chewing the test meal, while males showed little change, this could be partly due to the known relationship between estrogen and GH [12]. Further studies are required to establish whether increased mastication during meals can contribute to long-term GH increase and muscle mass.

## 5. Patents

This work has a patent applied (2022-069365), which can be accessed at https://www.j-platpat.inpit.go.jp/c1800/PU/JP-2022-069365/029B6CFB881BCB315636362F0886A90E8CB7739D7650EC62376F8705FDFE8FD0/11/ja (accessed on 1 July 2023)

## Figures and Tables

**Figure 1 nutrients-15-03628-f001:**
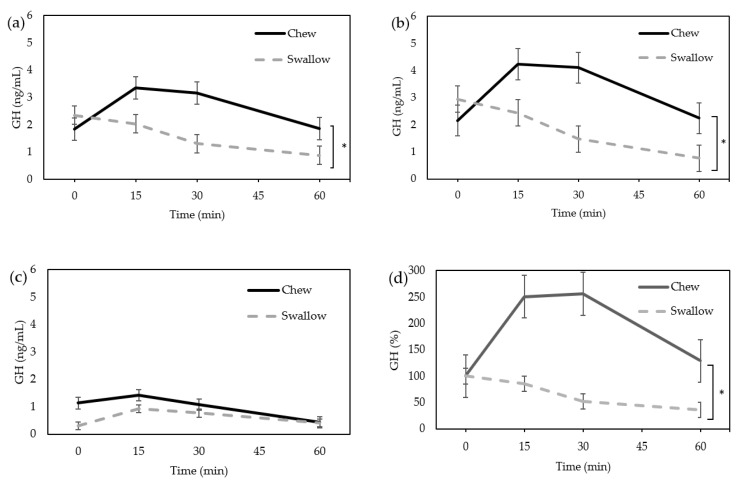
Average change of serum GH concentration. (**a**) Total (n = 13); (**b**) female only (n = 10); (**c**) male only (n = 3); (**d**) the percentage increase in serum GH in females. * Statistically significant differences between groups.

**Figure 2 nutrients-15-03628-f002:**
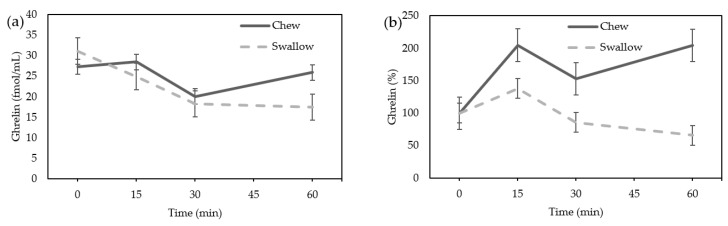
(**a**) Average change of acylated ghrelin concentration in females; (**b**) the percentage increase of ghrelin in females.

**Figure 3 nutrients-15-03628-f003:**
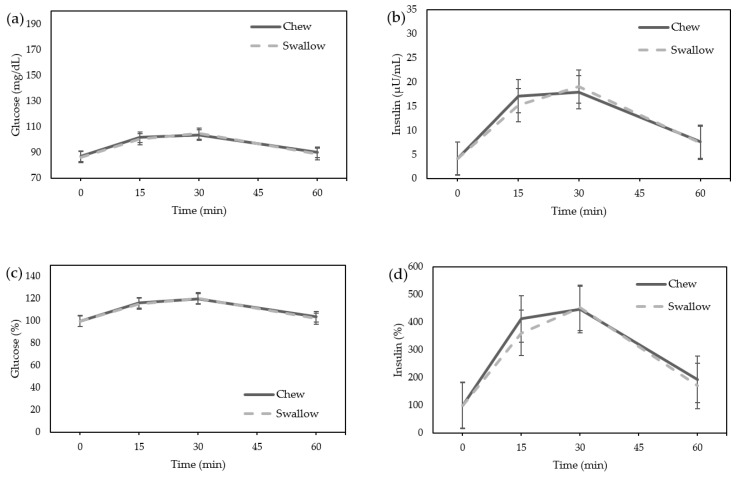
Average change in glucose and insulin concentrations in females. (**a**) Glucose; (**b**) Insulin; The percentage increase of glucose and insulin in females. (**c**) Glucose; (**d**) Insulin.

**Figure 4 nutrients-15-03628-f004:**
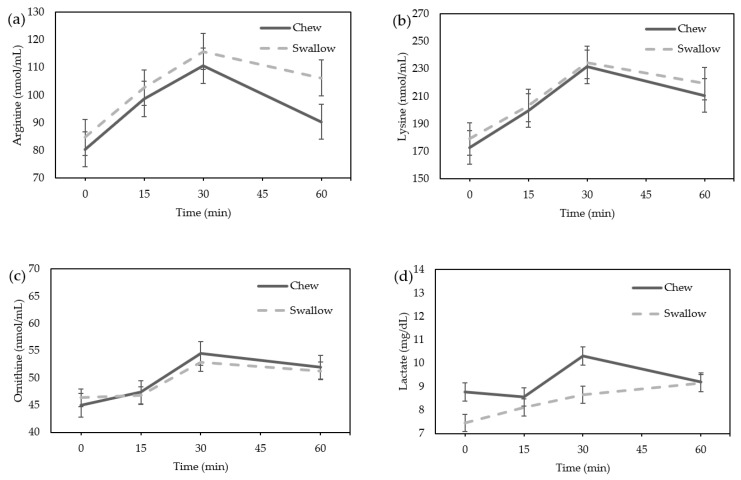
Average change in amino acids and lactate concentrations in females: (**a**) arginine; (**b**) lysine; (**c**) ornithine; (**d**) lactate concentration; the percentage increase in amino acids and lactate in females. (**e**) Arginine; (**f**) lysine; (**g**) ornithine; (**h**) lactate.

## Data Availability

The data presented in this study are available on request from the corresponding author.

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
