# Peer review of "Augmentation of Growth Hormone by Chewing in Females"

_nutrients, 2023, doi:10.3390/nu15163628_

Round 1
Reviewer 1 Report
Overview
The manuscript by Okamura et al. explored the effects of chewing a small nutrition bar on circulating hormone levels related to satiety and anabolism (e.g., GH, insulin). It is notable to tackle an area with widespread performance and health implications. The authors note some interesting findings that add to the growing body of literature. However, the authors should address many notable concerns. Specific comments are outlined below:
Comments
Major
Overall: The introduction does not set up the aim well and the authors do not report a hypothesis. It is relatively unclear how mastication promotes ghrelin and/or GH release. Furthermore, it is not made clear if mastication promotes hormone release and/or leads to increase circulating concentrations (as these are not the same thing). The introduction, in its current state, is not focused and needs to be addressed.
Overall: What is meant by “Due to the differences in GH secretion between males and females”? (lines 142-143). If there is an inherent sex difference in GH secretion, this needs to be addressed in the introduction, controlled for in the methodology, and the two sexes should not be combined in any reporting of data.
Overall: Many statements are made about outcomes peaking at various times. Unless statistics with post-hoc analysis were run to defend these statements about values being different at two or more time points, those statements/comparisons cannot be made. If the authors wish to identify peaks in the timecourse of analyses conducted, post-hoc comparisons are needed to be made.
Abstract: Please add more meaningful rationale for the study in the Abstract.
Abstract: Timing of blood sample collection?
Abstract: Other macronutrients contained in nutrition bar?
Abstract: Note magnitude of change (pre/post concentrations, % change) and relevant P value
Introduction: Commentary on the relationship between exercise and GH does not seem relevant to the current study.
Introduction: It is not clear if the authors are reporting arginine intake increases GH release. Does the route/form arginine or other consumed AA’s appear in for leading to GH release (i..e, solid vs liquid, plant vs animal derived, with or without mastication)? What quantity of these AA’s need to be taken to induce GH release? To what degree does GH change following AA intake? Please expand on these areas.
Introduction: How is glucose decreasing GH levels relevant where that statement occurs? This comment is unexpected and does not relate to the discussion in which it appears.
Introduction: What is the relationship between non- and acylated ghrelin? Can these be converted to each other? If so, how?
Methods: Please provide more detail about when medications were abstained (i.e., during the trial, in the day leading up to the trial, in the week leading up to the trial, etc)
Methods: What is meant by “individually allocated” (line 83)? Were participants randomized in any way?
Methods: Were participants allowed to consume caffeine the day of the trials?
Methods: Were any participants on oral contraceptives and/or post-menopausal? (May impact estrogen and GH levels)
Methods: Was exercise abstained from in the 24-48h prior to each trial?
Methods: Why was this bar specifically selected? Please provide some rational for this choice of nutrition.
Methods: Why/how was it decided that participants should chew the bar 300 times? Please provide rationale.
Methods: Please provide more product details for measurements/ELISA’s conducted (product item #, etc).
Methods: Please provide more detail about how plasma AA’s were fractionated via HPLC and/or add relevant references to support methodology.
Methods: Was a post-hoc analysis performed to explore noted differences? If so, which?
Results: Was the difference in GH between the groups found a main effect of time? The authors are encouraged to calculate the area under the curve for each group and compare via ttest as another metric for comparison.
Results: There appears to be an outlier in Figure 2b. The authors are encouraged to develop a strategy to remove statistical outliers when relevant and re-interpret the data. These outliers will skew findings and lead to misinterpretations.
Discussion: The statement about GH increasing from 15 to 60 minutes increases with chewing cannot be made. That statement is not statistically supported (i.e., no post-hoc analysis). Remove that statement or run the appropriate statistics. The same concept applies to all other related statements (e.g., ghrelin increasing at 15 minutes, etc).
Discussion: GH secretion was not assessed (e.g., line 179, etc). Changes in circulating GH levels were assessed. The authors need to adjust language to reflect the measurements that were actually made.
Discussion: The meal included a relatively high proportion of protein (34%). However, the bar was composed of only 5.56 grams of protein – a low protein load. Do the authors suspect the relative proportion of protein is more important than total protein as it relates to ghrelin and GH levels?
Discussion: The discussion section includes unexpected commentary on exercise, unrelated to the current study. Please remove exercise-related content as it does not relate to methodology conducted here.
Minor
The authors are suggested to label the two groups/trials with more clearly distinct terminology and be consistent with that terminology throughout (i.e., “Chew” vs “Swallow” or “Chew trial” vs “Swallow trial).
Which vein was blood collected from?
N/A
Reviewer 2 Report
The objective of the study is quite interesting, but this manuscript requires substantial revision.
Introduction – the logical way of the presentation should be as following;
1. The importance of GH in pathology (e.g., age-related sarcopenia)
2. List factors influencing GH secretion, starting from ghrelin, some amino acids, glucose, etc
3. Then list your research question and including those factors might influencing GH secretion
4. By following this presentation, it will guide the readers to a better understanding of your study design and workflow.
Results:
Quite confused as the way they present their results.
Figure 1, data were analyzed into total, females only and then males only
But for figure 2 to figure 4, are those for combined subjects or females only or males only? They need to explicitly provide the information in the figure legends.
I would prefer they analyze their data as in Figure 1, i.e., for total subjects, females, males
Further comments:
1. Very small study subjects are the number one concern. Can they increase the study subjects, esp males.
2. I may need to further comment on the revised manuscript (especially the pertaining discussion) once I read their revised figure 2 to figure 4.
3. Can they also measure serum gastrin level?
4. The two major molecular forms of circulating ghrelin are acyl ghrelin ( < 10 % ), which promotes food intake, and des-acyl ghrelin, which induces a negative energy balance. Can they measure serum concentration of des-acyl ghrelin?
Minor language editing is required.
Round 2
Reviewer 1 Report
This reviewer appreciates the substantial effort by the authors in revising the original manuscript. No further revisions are necessary.
Reviewer 2 Report
The authors have attempted to address most of the concerns raised in my previous comments.
As I've pointed out in the previous comments. Serum gastrin level as well as the complex interaction between Des-acyl gherlin vs acyl gherlin are the most critical confounding factors for this research. I fully understand that lack of funding as well as the absence of additional serum samples are some of the roadblocks to address those critical issues pertaining to this research. But at least, the authors ought to discuss those limitations in the section of discussion. The authors need to go a little deeper into the relevant molecular mechanisms as why and how chewing influences expression of those 3 molecules (gastrin, des-acyl gherlin vs acyl gherlin). It will give the readers a broad picture of the research theme.
minor concerns:
1. Abstract - 12203 ±15402 25 vs 3735 ± 988, unit? ng/mL?
2. Introduction and throughout the manuscript, pls make sure whether it refers to serum or plasma concentration/content?
e.g., page 2, line 83-84,
The aim of this study was to examine the effect of chew-83 ing on (serum, plasma or circulating??) GH and ghrelin levels using a nutrition bar.
3. We recruited the subjects who did not regularly take medication. The participants were instructed to abstain (from?) the medications as needed during the trial.
Language editing by a native user is recommended.
